# Sparse Overlapping Sets Lasso for Multitask Learning and its Application to fMRI Analysis

**Nikhil S. Rao**[†]
nrao2@wisc.edu

**Christopher R. Cox**[#]
crcox@wisc.edu

**Robert D. Nowak**[†]
nowak@ece.wisc.edu

**Timothy T. Rogers**[#]
ttrogers@wisc.edu

[†] **Department of Electrical and Computer Engineering,** [#] **Department of Psychology**
University of Wisconsin- Madison

## Abstract

Multitask learning can be effective when features useful in one task are also useful for other tasks, and the group lasso is a standard method for selecting a common subset of features. In this paper, we are interested in a less restrictive form of multitask learning, wherein (1) the available features can be organized into subsets according to a notion of similarity and (2) features useful in one task are similar, but not necessarily identical, to the features best suited for other tasks. The main contribution of this paper is a new procedure called *Sparse Overlapping Sets (SOS) lasso*, a convex optimization that automatically selects similar features for related learning tasks. Error bounds are derived for SOSlasso and its consistency is established for squared error loss. In particular, SOSlasso is motivated by multi-subject fMRI studies in which functional activity is classified using brain voxels as features. Experiments with real and synthetic data demonstrate the advantages of SOSlasso compared to the lasso and group lasso.

## 1   Introduction

Multitask learning exploits the relationships between several learning tasks in order to improve performance, which is especially useful if a common subset of features are useful for all tasks at hand. The group lasso (Glasso) [19, 8] is naturally suited for this situation: if a feature is selected for one task, then it is selected for all tasks. This may be too restrictive in many applications, and this motivates a less rigid approach to multitask feature selection. Suppose that the available features can be organized into overlapping subsets according to a notion of similarity, and that the features useful in one task are similar, but not necessarily identical, to those best suited for other tasks. In other words, a feature that is useful for one task suggests that the subset it belongs to may contain the features useful in other tasks (Figure 1).

In this paper, we introduce the *sparse overlapping sets lasso* (SOSlasso), a convex program to recover the sparsity patterns corresponding to the situations explained above. SOSlasso generalizes lasso [16] and Glasso, effectively spanning the range between these two well-known procedures. SOSlasso is capable of exploiting the similarities between useful features across tasks, but unlike Glasso it does not force different tasks to use exactly the same features. It produces sparse solutions, but unlike lasso it encourages similar patterns of sparsity across tasks. Sparse group lasso [14] is a special case of SOSlasso that only applies to disjoint sets, a significant limitation when features cannot be easily partitioned, as is the case of our motivating example in fMRI. The main contribution of this paper is a theoretical analysis of SOSlasso, which also covers sparse group lasso as a special case (further differentiating us from [14]). The performance of SOSlasso is analyzed, error

bounds are derived for general loss functions, and its consistency is shown for squared error loss. Experiments with real and synthetic data demonstrate the advantages of SOSlasso relative to lasso and Glasso.

## 1.1 Sparse Overlapping Sets

SOSlasso encourages sparsity patterns that are similar, but not identical, across tasks. This is accomplished by decomposing the features of each task into groups $G_1 \ldots G_M$, where $M$ is the same for each task, and $G_i$ is a set of features that can be considered similar across tasks. Conceptually, SOSlasso first selects subsets that are most useful for all tasks, and then identifies a unique sparse solution for each task drawing only from features in the selected subsets. In the fMRI application discussed later, the subsets are simply clusters of adjacent spatial data points (voxels) in the brains of multiple subjects. Figure 1 shows an example of the patterns that typically arise in sparse multitask learning applications, where rows indicate features and columns correspond to tasks.

Past work has focused on recovering variables that exhibit within and across group sparsity, when the groups do not overlap [14], finding application in genetics, handwritten character recognition [15] and climate and oceanography [2]. Along related lines, the exclusive lasso [21] can be used when it is explicitly known that variables in certain sets are negatively correlated.

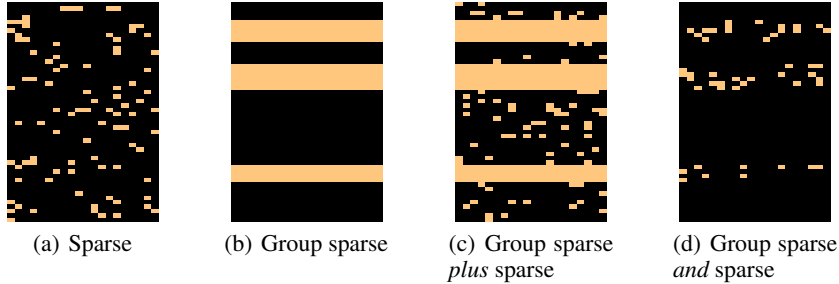

|         |                 |                         |                        |
|---------|-----------------|-------------------------|------------------------|
| (a) Sparse | (b) Group sparse | (c) Group sparse *plus* sparse | (d) Group sparse *and* sparse |

Figure 1: A comparison of different sparsity patterns. (a) shows a standard sparsity pattern. An example of group sparse patterns promoted by Glasso [19] is shown in (b). In (c), we show the patterns considered in [6]. Finally, in (d), we show the patterns we are interested in this paper.

## 1.2 fMRI Applications

In psychological studies involving fMRI, multiple participants are scanned while subjected to exactly the same experimental manipulations. Cognitive Neuroscientists are interested in identifying the patterns of activity associated with different cognitive states, and construct a model of the activity that accurately predicts the cognitive state evoked on novel trials. In these datasets, it is reasonable to expect that the same general areas of the brain will respond to the manipulation in every participant. However, the specific patterns of activity in these regions will vary, both because neural codes can vary by participant [4] and because brains vary in size and shape, rendering neuroanatomy only an approximate guide to the location of relevant information across individuals. In short, a voxel useful for prediction in one participant suggests the general anatomical neighborhood where useful voxels may be found, but not the precise voxel. While logistic Glasso [17], lasso [13], and the elastic net penalty [12] have been applied to neuroimaging data, these methods do not exclusively take into account both the common macrostructure and the differences in microstructure across brains. SOSlasso, in contrast, lends itself well to such a scenario, as we will see from our experiments.

## 1.3 Organization

The rest of the paper is organized as follows: in Section 2, we outline the notations that we will use and formally set up the problem. We also introduce the SOSlasso regularizer. We derive certain key properties of the regularizer in Section 3. In Section 4, we specialize the problem to the multitask linear regression setting (2), and derive consistency rates for the same, leveraging ideas from [9]. We outline experiments performed on simulated data in Section 5. In this section, we also perform logistic regression on fMRI data, and argue that the use of the SOSlasso yields interpretable multivariate solutions compared to Glasso and lasso.

## 2 Sparse Overlapping Sets Lasso

We formalize the notations used in the sequel. Lowercase and uppercase bold letters indicate vectors and matrices respectively. We assume a multitask learning framework, with a data matrix $\boldsymbol{\Phi}_t \in \mathbb{R}^{n \times p}$ for each task $t \in \{1, 2, \ldots, \mathcal{T}\}$. We assume there exists a vector $\boldsymbol{x}_t^\star \in \mathbb{R}^p$ such that measurements obtained are of the form $\boldsymbol{y}_t = \boldsymbol{\Phi}_t \boldsymbol{x}_t^\star + \eta_t \quad \eta_t \sim \mathcal{N}(0, \sigma^2 \boldsymbol{I})$. Let $\boldsymbol{X}^\star := [\boldsymbol{x}_1^\star \ \boldsymbol{x}_2^\star \ \ldots \boldsymbol{x}_{\mathcal{T}}^\star] \in \mathbb{R}^{p \times \mathcal{T}}$. Suppose we are given $M$ (possibly overlapping) groups $\tilde{\mathcal{G}} = \{\tilde{G}_1, \tilde{G}_2, \ldots, \tilde{G}_M\}$, so that $\tilde{G}_i \subset \{1, 2, \ldots, p\} \ \forall i$, of maximum size $B$. These groups contain sets of "similar" features, the notion of similarity being application dependent. We assume that all but $k \ll M$ groups are identically zero. Among the active groups, we further assume that at most only a fraction $\alpha \in (0, 1)$ of the coefficients per group are non zero. We consider the following optimization program in this paper

$$\hat{\boldsymbol{X}} = \arg\min_{\boldsymbol{x}} \left\{ \sum_{t=1}^{\mathcal{T}} \mathcal{L}_{\boldsymbol{\Phi}_t}(\boldsymbol{x}_t) + \lambda_n h(\boldsymbol{x}) \right\} \tag{1}$$

where $\boldsymbol{x} = [\boldsymbol{x}_1^T \boldsymbol{x}_2^T \ldots \boldsymbol{x}_{\mathcal{T}}^T]^T$, $h(\boldsymbol{x})$ is a regularizer and $\mathcal{L}_t := \mathcal{L}_{\boldsymbol{\Phi}_t}(\boldsymbol{x}_t)$ denotes the loss function, whose value depends on the data matrix $\boldsymbol{\Phi}_t$. We consider least squares and logistic loss functions. In the least squares setting, we have $\mathcal{L}_t = \frac{1}{2n}\|\boldsymbol{y}_t - \boldsymbol{\Phi}_t \boldsymbol{x}_t\|^2$. We reformulate the optimization problem (1) with the least squares loss as

$$\hat{\boldsymbol{x}} = \arg\min_{\boldsymbol{x}} \left\{ \frac{1}{2n}\|\boldsymbol{y} - \boldsymbol{\Phi}\boldsymbol{x}\|_2^2 + \lambda_n h(\boldsymbol{x}) \right\} \tag{2}$$

where $\boldsymbol{y} = [\boldsymbol{y}_1^T \boldsymbol{y}_2^T \ldots \boldsymbol{y}_{\mathcal{T}}^T]^T$ and the block diagonal matrix $\boldsymbol{\Phi}$ is formed by block concatenating the $\boldsymbol{\Phi}_t's$. We use this reformulation for ease of exposition (see also [8] and references therein). Note that $\boldsymbol{x} \in \mathbb{R}^{\mathcal{T}p}$, $\boldsymbol{y} \in \mathbb{R}^{\mathcal{T}n}$, and $\boldsymbol{\Phi} \in \mathbb{R}^{\mathcal{T}n \times \mathcal{T}p}$. We also define $\mathcal{G} = \{G_1, G_2, \ldots, G_M\}$ to be the set of groups defined on $\mathbb{R}^{\mathcal{T}p}$ formed by aggregating the rows of $\boldsymbol{X}$ that were originally in $\tilde{\mathcal{G}}$, so that $\boldsymbol{x}$ is composed of groups $G \in \mathcal{G}$.

We next define a regularizer $h$ that promotes sparsity both within and across overlapping sets of similar features:

$$h(\boldsymbol{x}) = \inf_{\mathcal{W}} \sum_{G \in \mathcal{G}} (\alpha_G \|\boldsymbol{w}_G\|_2 + \|\boldsymbol{w}_G\|_1) \quad \textbf{s.t.} \quad \sum_{G \in \mathcal{G}} \boldsymbol{w}_G = \boldsymbol{x} \tag{3}$$

where the $\alpha_G > 0$ are constants that balance the tradeoff between the group norms and the $\ell_1$ norm. Each $\boldsymbol{w}_G$ has the same size as $\boldsymbol{x}$, with support restricted to the variables indexed by group $G$. $\mathcal{W}$ is a set of vectors, where each vector has a support restricted to one of the groups $G \in \mathcal{G}$:

$$\mathcal{W} = \{\boldsymbol{w}_G \in \mathbb{R}^{\mathcal{T}p}| \ [\boldsymbol{w}_G]_i = 0 \text{ if } i \notin G\}$$

where $[\boldsymbol{w}_G]_i$ is the $i^{th}$ coefficient of $\boldsymbol{w}_G$. The *SOSlasso* is the optimization in (1) with $h(\boldsymbol{x})$ as defined in (3).

We say the set of vectors $\boldsymbol{w}_G$ is an optimal decomposition of $\boldsymbol{x}$ if they achieve the inf in (3). The objective function in (3) is convex and coercive. Hence, $\forall \boldsymbol{x}$, an optimal decomposition always exists.

As the $\alpha_G \to \infty$ the $\ell_1$ term becomes redundant, reducing $h(\boldsymbol{x})$ to the overlapping group lasso penalty introduced in [5], and studied in [10, 11]. When the $\alpha_G \to 0$, the overlapping group lasso term vanishes and $h(\boldsymbol{x})$ reduces to the lasso penalty. We consider $\alpha_G = 1 \ \forall G$. All the results in the paper can be easily modified to incorporate different settings for the $\alpha_G$.

| Support | Values | $\sum_G \|\boldsymbol{x}_G\|_2$ | $\|\boldsymbol{x}\|_1$ | $\sum_G (\|\boldsymbol{x}_G\|_2 + \|\boldsymbol{x}_G\|_1)$ |
|---|---|---|---|---|
| $\{1, 4, 9\}$ | $\{3, 4, 7\}$ | 12 | 14 | 26 |
| $\{1, 2, 3, 4, 5\}$ | $\{2, 5, 2, 4, 5\}$ | 8.602 | 18 | 26.602 |
| $\{1, 3, 4\}$ | $\{3, 4, 7\}$ | 8.602 | 14 | 22.602 |

Table 1: Different instances of a 10-d vector and their corresponding norms.

The example in Table 1 gives an insight into the kind of sparsity patterns preferred by the function $h(\boldsymbol{x})$. The optimization problems (1) and (2) will prefer solutions that have a small value of $h(\cdot)$.

Consider 3 instances of $\boldsymbol{x} \in \mathbb{R}^{10}$, and the corresponding group lasso, $\ell_1$, and $h(\boldsymbol{x})$ function values. The vector is assumed to be made up of two groups, $G_1 = \{1, 2, 3, 4, 5\}$ and $G_2 = \{6, 7, 8, 9, 10\}$. $h(\boldsymbol{x})$ is smallest when the support set is sparse within groups, and also when only one of the two groups is selected. The $\ell_1$ norm does not take into account sparsity across groups, while the group lasso norm does not take into account sparsity within groups.

To solve (1) and (2) with the regularizer proposed in (3), we use the covariate duplication method of [5], to reduce the problem to a non overlapping sparse group lasso problem. We then use proximal point methods [7] in conjunction with the MALSAR [20] package to solve the optimization problem.

## 3 Error Bounds for SOSlasso with General Loss Functions

We derive certain key properties of the regularizer $h(\cdot)$ in (3), independent of the loss function used.

**Lemma 3.1** *The function $h(\boldsymbol{x})$ in (3) is a norm*

The proof follows from basic properties of norms and because if $\boldsymbol{w}_G, \boldsymbol{v}_G$ are optimal decompositions of $\boldsymbol{x}, \boldsymbol{y}$, then it does not imply that $\boldsymbol{w}_G + \boldsymbol{v}_G$ is an optimal decomposition of $\boldsymbol{x} + \boldsymbol{y}$. For a detailed proof, please refer to the supplementary material.

The dual norm of $h(\boldsymbol{x})$ can be bounded as

$$
\begin{aligned}
h^*(\boldsymbol{u}) &= \max_{\boldsymbol{x}} \{\boldsymbol{x}^T \boldsymbol{u}\} \ \ \textbf{s.t.} \ \ h(\boldsymbol{x}) \le 1 \\
&= \max_{\mathcal{W}} \{\sum_{G \in \mathcal{G}} \boldsymbol{w}_G^T \boldsymbol{u}_G\} \ \ \textbf{s.t.} \ \ \sum_{G \in \mathcal{G}} (\|\boldsymbol{w}_G\|_2 + \|\boldsymbol{w}_G\|_1) \le 1 \\
&\overset{(i)}{\le} \max_{\mathcal{W}} \{\sum_{G \in \mathcal{G}} \boldsymbol{w}_G^T \boldsymbol{u}_G\} \ \ \textbf{s.t.} \ \ \sum_{G \in \mathcal{G}} 2\|\boldsymbol{w}_G\|_2 \le 1 \\
&= \max_{\mathcal{W}} \{\sum_{G \in \mathcal{G}} \boldsymbol{w}_G^T \boldsymbol{u}_G\} \ \ \textbf{s.t.} \ \ \sum_{G \in \mathcal{G}} \|\boldsymbol{w}_G\|_2 \le \frac{1}{2} \\
\Rightarrow h^*(\boldsymbol{u}) &\le \max_{G \in \mathcal{G}} \frac{1}{2}\|\boldsymbol{u}_G\|_2
\end{aligned}
\tag{4}
$$

(i) follows from the fact that the constraint set in (i) is a superset of the constraint set in the previous statement, since $\|\boldsymbol{a}\|_2 \le \|\boldsymbol{a}\|_1$. (4) follows from noting that the maximum is obtained by setting $\boldsymbol{w}_{G^*} = \frac{\boldsymbol{u}_{G^*}}{2\|\boldsymbol{u}_{G^*}\|_2}$, where $G^* = \arg\max_{G \in \mathcal{G}} \|\boldsymbol{u}_G\|_2$. The inequality (4) is far more tractable than the actual dual norm, and will be useful in our derivations below. Since $h(\cdot)$ is a norm, we can apply methods developed in [9] to derive consistency rates for the optimization problems (1) and (2). We will use the same notations as in [9] wherever possible.

**Definition 3.2** *A norm $h(\cdot)$ is decomposable with respect to the subspace pair $sA \subset sB$ if $h(\boldsymbol{a} + \boldsymbol{b}) = h(\boldsymbol{a}) + h(\boldsymbol{b}) \ \ \forall \boldsymbol{a} \in sA, \boldsymbol{b} \in sB^\perp$.*

**Lemma 3.3** *Let $\boldsymbol{x}^\star \in \mathbb{R}^p$ be a vector that can be decomposed into (overlapping) groups with within-group sparsity. Let $\mathcal{G}^\star \subset \mathcal{G}$ be the set of active groups of $\boldsymbol{x}^\star$. Let $S = supp(\boldsymbol{x}^\star)$ indicate the support set of $\boldsymbol{x}$. Let $sA$ be the subspace spanned by the coordinates indexed by $S$, and let $sB = sA$. We then have that the norm in (3) is decomposable with respect to $sA, sB$*

The result follows in a straightforward way from noting that supports of decompositions for vectors in $sA$ and $sB^\perp$ do not overlap. We defer the proof to the supplementary material.

**Definition 3.4** *Given a subspace $sB$, the subspace compatibility constant with respect to a norm $\| \ \|$ is given by*

$$
\Psi(B) = \sup \left\{ \frac{h(\boldsymbol{x})}{\|\boldsymbol{x}\|} \ \ \forall \boldsymbol{x} \in sB \backslash \{\boldsymbol{0}\} \right\}
$$

**Lemma 3.5** *Consider a vector $\boldsymbol{x}$ that can be decomposed into $\mathcal{G}^\star \subset \mathcal{G}$ active groups. Suppose the maximum group size is $B$, and also assume that a fraction $\alpha \in (0, 1)$ of the coordinates in each active group is non zero. Then,*

$$
h(\boldsymbol{x}) \le (1 + \sqrt{B\alpha})\sqrt{|\mathcal{G}^\star|}\|\boldsymbol{x}\|_2
$$

**Proof** For any vector $\boldsymbol{x}$ with $supp(\boldsymbol{x}) \subset \mathcal{G}^\star$, there exists a representation $\boldsymbol{x} = \sum_{G \in \mathcal{G}^\star} \boldsymbol{w}_G$, such that the supports of the different $\boldsymbol{w}_G$ do not overlap. Then,

$$h(\boldsymbol{x}) \leq \sum_{G \in \mathcal{G}^\star} (\|\boldsymbol{w}_G\|_2 + \|\boldsymbol{w}_G\|_1) \ \leq (1 + \sqrt{B\alpha}) \sum_{G \in \mathcal{G}^\star} \|\boldsymbol{w}_G\|_2 \ \leq (1 + \sqrt{B\alpha})\sqrt{|\mathcal{G}^\star|}\|\boldsymbol{x}\|_2$$

∎

We see that $(1 + \sqrt{B\alpha})\sqrt{|\mathcal{G}^\star|}$ (Lemma 3.5) gives an upper bound on the subspace compatibility constant with respect to the $\ell_2$ norm for the subspace indexed by the support of the vector, which is contained in the span of the union of groups in $\mathcal{G}^\star$.

**Definition 3.6** *For a given set $S$, and given vector $\boldsymbol{x}^\star$, the loss function $\mathcal{L}_{\boldsymbol{\Phi}}(\boldsymbol{x})$ satisfies the Restricted Strong Convexity(RSC) condition with parameter $\kappa$ and tolerance $\tau$ if*

$$\mathcal{L}_{\boldsymbol{\Phi}}(\boldsymbol{x}^\star + \Delta) - \mathcal{L}_{\boldsymbol{\Phi}}(\boldsymbol{x}^\star) - \langle \nabla \mathcal{L}_{\boldsymbol{\Phi}}(\boldsymbol{x}^\star), \Delta \rangle \geq \kappa\|\Delta\|_2^2 - \tau^2(\boldsymbol{x}^\star) \ \ \forall \Delta \in S$$

In this paper, we consider vectors $\boldsymbol{x}^\star$ that lie *exactly* in $k \ll M$ groups, and display within-group sparsity. This implies that the tolerance $\tau(\boldsymbol{x}^\star) = 0$, and we will ignore this term henceforth.

We also define the following set, which will be used in the sequel:

$$C(sA, sB, \boldsymbol{x}^\star) := \{\Delta \in \mathbb{R}^p | h(\Pi_{sB^\perp}\Delta) \leq 3h(\Pi_{sB}\Delta) + 4h(\Pi_{sA^\perp}\boldsymbol{x}^\star)\} \tag{5}$$

where $\Pi_{sA}(\cdot)$ denotes the projection onto the subspace $sA$. Based on the results above, we can now apply a result from [9] to the SOSlasso:

**Theorem 3.7** *(Corollary 1 in [9]) Consider a convex and differentiable loss function such that RSC holds with constants $\kappa$ and $\tau = 0$ over (5), and a norm $h(\cdot)$ decomposable over sets $sA$ and $sB$. For the optimization program in (1), using the parameter $\lambda_n \geq 2h^*(\nabla \mathcal{L}_{\boldsymbol{\Phi}}(\boldsymbol{x}^\star))$, any optimal solution $\hat{\boldsymbol{x}}_{\lambda_n}$ to (1) satisfies*

$$\|\hat{\boldsymbol{x}}_{\lambda_n} - \boldsymbol{x}^\star\|_2 \leq \frac{9\lambda_n^2}{\kappa}\Psi^2(sB)$$

The result above shows a general bound on the error using the lasso with sparse overlapping sets. Note that the regularization parameter $\lambda_n$ as well as the RSC constant $\kappa$ depend on the loss function $\mathcal{L}_{\boldsymbol{\Phi}}(\boldsymbol{x})$. Convergence for logistic regression settings may be derived using methods in [1]. In the next section, we consider the least squares loss (2), and show that the estimate using the SOSlasso is consistent.

## 4 Consistency of SOSlasso with Squared Error Loss

We first need to bound the dual norm of the gradient of the loss function, so as to bound $\lambda_n$. Consider $\mathcal{L} := \mathcal{L}_{\boldsymbol{\Phi}}(\boldsymbol{x}) = \frac{1}{2n}\|\boldsymbol{y} - \boldsymbol{\Phi}\boldsymbol{x}\|^2$. The gradient of the loss function with respect to $\boldsymbol{x}$ is given by $\nabla \mathcal{L} = \frac{1}{n}\boldsymbol{\Phi}^T(\boldsymbol{\Phi}\boldsymbol{x} - \boldsymbol{y}) = \frac{1}{n}\boldsymbol{\Phi}^T\eta$ where $\eta = [\eta_1^T \eta_2^T \ldots \eta_T^T]^T$ (see Section 2). Our goal now is to find an upper bound on the quantity $h^*(\nabla \mathcal{L})$, which from (4) is

$$\frac{1}{2}\max_{G \in \mathcal{G}} \|\nabla \mathcal{L}_G\|_2 = \frac{1}{2n}\max_{G \in \mathcal{G}} \|\boldsymbol{\Phi}_G^T \eta\|_2$$

where $\boldsymbol{\Phi}_G$ is the matrix $\boldsymbol{\Phi}$ restricted to the columns indexed by the group $G$. We will prove an upper bound for the above quantity in the course of the results that follow.

Since $\eta \sim \mathcal{N}(0, \sigma^2 \boldsymbol{I})$, we have $\boldsymbol{\Phi}_G^T \eta \sim \sigma \mathcal{N}(0, \boldsymbol{\Phi}_G^T \boldsymbol{\Phi}_G)$. Defining $\sigma_{mG} := \sigma_{\max}\{\boldsymbol{\Phi}_G^T \boldsymbol{\Phi}_G\}$ to be the maximum singular value, we have $\|\boldsymbol{\Phi}_G^T \eta\|_2^2 \leq \sigma^2 \sigma_{mG}^2 \|\gamma\|_2^2$, where $\gamma \sim \mathcal{N}(0, \boldsymbol{I}_{|G|}) \Rightarrow \|\gamma\|_2^2 \sim \chi_{|G|}^2$, where $\chi_d^2$ is a chi-squared random variable with $d$ degrees of freedom. This allows us to work with the more tractable chi squared random variable when we look to bound the dual norm of $\nabla \mathcal{L}$. The next lemma helps us obtain a bound on the maximum of $\chi^2$ random variables.

**Lemma 4.1** *Let $\boldsymbol{z}_1, \boldsymbol{z}_2, \ldots, \boldsymbol{z}_M$ be chi-squared random variables with $d$ degrees of freedom. Then for some constant $c$,*

$$\mathbb{P}\left(\max_{i=1,2,\ldots,M} z_i \leq c^2 d\right) \geq 1 - \exp\left(\log(M) - \frac{(c-1)^2 d}{2}\right)$$

**Proof** From the chi-squared tail bound in [3], $\mathbb{P}(z_i \geq c^2 d) \leq \exp\left(-\frac{(c-1)^2 d}{2}\right)$. The result follows from a union bound and inverting the expression. ∎

**Lemma 4.2** *Consider the loss function $\mathcal{L} := \frac{1}{2n}\sum_{t=1}^{\mathcal{T}}\|\boldsymbol{y}_t - \boldsymbol{\Phi}_t \boldsymbol{x}_t\|^2 = \frac{1}{2n}\|\boldsymbol{y} - \boldsymbol{\Phi}\boldsymbol{x}\|^2$, with the $\boldsymbol{\Phi}_t's$ deterministic and the measurements corrupted with AWGN of variance $\sigma^2$. For the regularizer in (3), the dual norm of the gradient of the loss function is bounded as*

$$h^*(\nabla\mathcal{L})^2 \leq \frac{\sigma^2 \sigma_m^2}{4}\frac{(\log(M) + \mathcal{T}B)}{n}$$

*with probability at least $1 - c_1 \exp(-c_2 n)$, for $c_1, c_2 > 0$, and where $\sigma_m = \max_{G \in \mathcal{G}} \sigma_{mG}$*

**Proof** Let $\gamma \sim \chi^2_{\mathcal{T}|G|}$. We begin with the upper bound obtained for the dual norm of the regularizer in (4):

$$h^*(\nabla\mathcal{L})^2 \overset{(i)}{\leq} \frac{1}{4}\max_{G \in \mathcal{G}}\left\|\frac{1}{n}\boldsymbol{\Phi}_G^T \eta\right\|_2^2 \leq \frac{\sigma^2}{4}\max_{G \in \mathcal{G}}\frac{\sigma_{mG}^2 \gamma}{n^2}$$

$$\overset{(ii)}{\leq} \frac{\sigma^2 \sigma_m^2}{4}\max_{G \in \mathcal{G}}\frac{\gamma}{n^2} \overset{(iii)}{\leq} \frac{\sigma^2 \sigma_m^2}{4}c^2 \mathcal{T}B \quad \textbf{w. p.} \quad 1 - \exp\left(\log(M) - \frac{(cn-1)^2 \mathcal{T}B}{2}\right)$$

where $(i)$ follows from the formulation of the gradient of the loss function and the fact that the square of maximum of non negative numbers is the maximum of the squares of the same numbers. In $(ii)$, we have defined $\sigma_m = \max_G \sigma_{mG}$. Finally, we have made use of Lemma 4.1 in $(iii)$. We then set

$$c^2 = \frac{\log(M) + \mathcal{T}B}{\mathcal{T}Bn}$$

to obtain the result. ∎

We combine the results developed so far to derive the following consistency result for the SOS lasso, with the least squares loss function.

**Theorem 4.3** *Suppose we obtain linear measurements of a sparse overlapping grouped matrix $\boldsymbol{X}^\star \in \mathbb{R}^{p \times \mathcal{T}}$, corrupted by AWGN of variance $\sigma^2$. Suppose the matrix $\boldsymbol{X}^\star$ can be decomposed into $M$ possible overlapping groups of maximum size $B$, out of which $k$ are active. Furthermore, assume that a fraction $\alpha \in (0, 1]$ of the coefficients are non zero in each active group. Consider the following vectorized SOSlasso multitask regression problem (2):*

$$\widehat{\boldsymbol{x}} = \arg\min_{\boldsymbol{x}}\left\{\frac{1}{2n}\|\boldsymbol{y} - \boldsymbol{\Phi}\boldsymbol{x}\|_2^2 + \lambda_n h(\boldsymbol{x})\right\},$$

$$h(\boldsymbol{x}) = \inf_{\mathcal{W}}\sum_{G \in \mathcal{G}}(\|\boldsymbol{w}_G\|_2 + \|\boldsymbol{w}_G\|_1) \quad \textit{s.t.} \quad \sum_{G \in \mathcal{G}}\boldsymbol{w}_G = \boldsymbol{x}$$

*Suppose the data matrices $\boldsymbol{\Phi}_t$ are non random, and the loss function satisfies restricted strong convexity assumptions with parameter $\kappa$. Then, for $\lambda_n \geq \frac{\sigma^2 \sigma_m^2 (\log(M) + \mathcal{T}B)}{4n}$, the following holds with probability at least $1 - c_1 \exp(-c_2 n)$, with $c_1, c_2 > 0$:*

$$\|\widehat{\boldsymbol{x}} - \boldsymbol{x}^\star\|_2 \leq \frac{9}{4}\frac{\sigma^2 \sigma_m^2 \left(1 + \sqrt{\mathcal{T}B\alpha}\right)^2 k(\log(M) + \mathcal{T}B)}{n\kappa}$$

*where we define $\sigma_m := \max_{G \in \mathcal{G}}\sigma_{max}\{\boldsymbol{\Phi}_G^T \boldsymbol{\Phi}_G\}$*

**Proof** Follows from substituting in Theorem 3.7 the results from Lemma 3.5 and Lemma 4.2. ∎

From [9], we see that the convergence rate matches that of the group lasso, with an additional multiplicative factor $\alpha$. This stems from the fact that the signal has a sparse structure "embedded" within a group sparse structure. Visualizing the optimization problem as that of solving a lasso within a group lasso framework lends some intuition into this result. Note that since $\alpha < 1$, this bound is much smaller than that of the standard group lasso.

# 5 Experiments and Results

## 5.1 Synthetic data, Gaussian Linear Regression

For $\mathcal{T} = 20$ tasks, we define a $N = 2002$ element vector divided into $M = 500$ groups of size $B = 6$. Each group overlaps with its neighboring groups ($G_1 = \{1, 2, \ldots, 6\}$, $G_2 = \{5, 6, \ldots, 10\}$, $G_3 = \{9, 10, \ldots, 14\}$, ...). 20 of these groups were activated uniformly at random, and populated from a uniform $[-1, 1]$ distribution. A proportion $\alpha$ of these coefficients with largest magnitude were retained as true signal. For each task, we obtain 250 linear measurements using a $\mathcal{N}(0, \frac{1}{250}\boldsymbol{I})$ matrix. We then corrupt each measurement with Additive White Gaussian Noise (AWGN), and assess signal recovery in terms of Mean Squared Error (MSE). The regularization parameter was clairvoyantly picked to minimize the MSE over a range of parameter values. The results of applying lasso, standard latent group lasso [5, 10], and our SOSlasso to these data are plotted in Figures 2(a), varying $\sigma$, $\alpha = 0.2$, and 2(b), varying $\alpha$, $\sigma = 0.1$. Each point in Figures 2(a) and 2(b), is the average of 100 trials, where each trial is based on a new random instance of $\boldsymbol{X}^{\star}$ and the Gaussian data matrices.

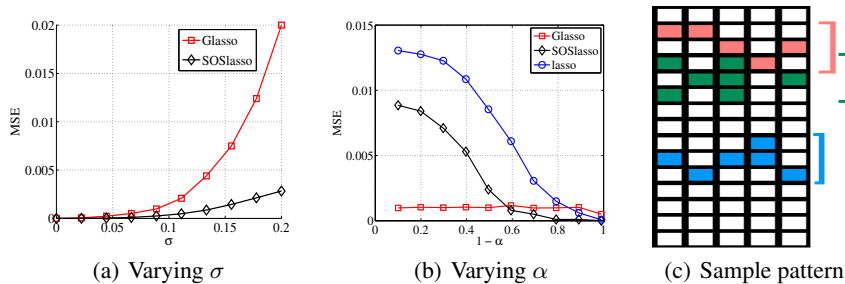

| (a) Varying $\sigma$ | (b) Varying $\alpha$ | (c) Sample pattern |

Figure 2: As the noise is increased (a), our proposed penalty function (SOSlasso) allows us to recover the true coefficients more accurately than the group lasso (Glasso). Also, when alpha is large, the active groups are not sparse, and the standard overlapping group lasso outperforms the other methods. However, as $\alpha$ reduces, the method we propose outperforms the group lasso (b). (c) shows a toy sparsity pattern, with different colors denoting different overlapping groups

## 5.2 The SOSlasso for fMRI

In this experiment, we compared SOSlasso, lasso, and Glasso in analysis of the star-plus dataset [18]. 6 subjects made judgements that involved processing 40 sentences and 40 pictures while their brains were scanned in half second intervals using fMRI[1]. We retained the 16 time points following each stimulus, yielding 1280 measurements at each voxel. The task is to distinguish, at each point in time, which stimulus a subject was processing. [18] showed that there exists cross-subject consistency in the cortical regions useful for prediction in this task. Specifically, experts partitioned each dataset into 24 non overlapping regions of interest (ROIs), then reduced the data by discarding all but 7 ROIs and, for each subject, averaging the BOLD response across voxels within each ROI and showed that a classifier trained on data from 5 subjects generalized when applied to data from a 6th.

We assessed whether SOSlasso could leverage this cross-individual consistency to aid in the discovery of predictive voxels without requiring expert pre-selection of ROIs, or data reduction, or any alignment of voxels beyond that existing in the raw data. Note that, unlike [18], we do not aim to learn a solution that generalizes to a withheld subject. Rather, we aim to discover a group sparsity pattern that suggests a similar set of voxels in all subjects, before optimizing a separate solution for each individual. If SOSlasso can exploit cross-individual anatomical similarity from this raw, coarsely-aligned data, it should show reduced cross-validation error relative to the lasso applied separately to each individual. If the solution is sparse within groups and highly variable across individuals, SOSlasso should show reduced cross-validation error relative to Glasso. Finally, if SOSlasso is finding useful cross-individual structure, the features it selects should align at least somewhat with the expert-identified ROIs shown by [18] to carry consistent information.

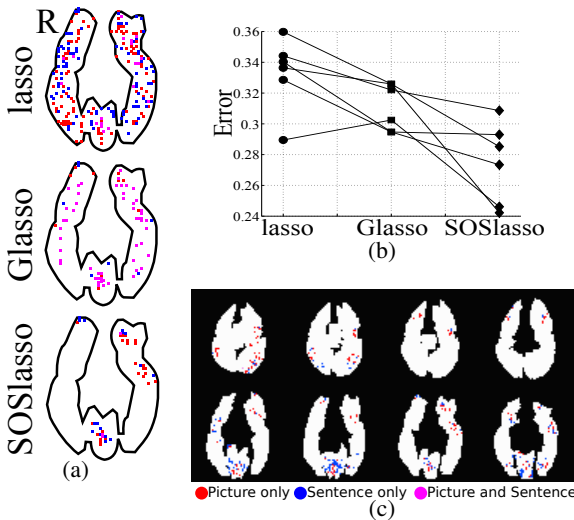

(a)

(b)

(c)

● Picture only ● Sentence only ● Picture and Sentence

Figure 3: Results from fMRI experiments. (a) Aggregated sparsity patterns for a single brain slice. (b) Cross-validation error obtained with each method. Lines connect data for a single subject. (c) The full sparsity pattern obtained with SOSlasso.

| Method | % ROI | t(5) , p |
|---|---|---|
| lasso | 46.11 | 6.08 ,0.001 |
| Glasso | 50.89 | 5.65 ,0.002 |
| SOSlasso | 70.31 | |

Table 2: Proportion of selected voxels in the 7 relevant ROIS aggregated over subjects, and corresponding two-tailed significance levels for the contrast of lasso and Glasso to SOSlasso.

We trained 3 classifiers using 4-fold cross validation to select the regularization parameter, considering all available voxels without preselection. We group regions of $5 \times 5 \times 1$ voxels and considered overlapping groups "shifted" by 2 voxels in the first 2 dimensions.[2] Figure 3(b) shows the individual error rates across the 6 subjects for the three methods. Across subjects, SOSlasso had a significantly lower cross-validation error rate (27.47 %) than individual lasso (33.3 %; within-subjects t(5) = 4.8; p = 0.004 two-tailed), showing that the method can exploit anatomical similarity across subjects to learn a better classifier for each. SOSlasso also showed significantly lower error rates than glasso (31.1 %; t(5) = 2.92; p = 0.03 two-tailed), suggesting that the signal is sparse within selected regions and variable across subjects.

Figure 3(a) presents a sample of the the sparsity patterns obtained from the different methods, aggregated over all subjects. Red points indicate voxels that contributed positively to picture classification in at least one subject, but never to sentences; Blue points have the opposite interpretation. Purple points indicate voxels that contributed positively to picture and sentence classification in different subjects. The remaining slices for the SOSlasso are shown in Figure 3(c). There are three things to note from Figure 3(a). First, the Glasso solution is fairly dense, with many voxels signaling both picture and sentence across subjects. We believe this "purple haze" demonstrates why Glasso is ill-suited for fMRI analysis: a voxel selected for one subject must also be selected for all others. This approach will not succeed if, as is likely, there exists no direct voxel-to-voxel correspondence or if the neural code is variable across subjects. Second, the lasso solution is less sparse than the SOSlasso because it allows any task-correlated voxel to be selected. It leads to a higher cross-validation error, indicating that the ungrouped voxels are inferior predictors (Figure 3(b)). Third, the SOSlasso not only yields a sparse solution, but also clustered. To assess how well these clusters align with the anatomical regions thought *a-priori* to be involved in sentence and picture representation, we calculated the proportion of selected voxels falling within the 7 ROIs identified by [18] as relevant to the classification task (Table 2). For SOSlasso an average of 70% of identified voxels fell within these ROIs, significantly more than for lasso or Glasso.

## 6   Conclusions and Extensions

We have introduced SOSlasso, a function that recovers sparsity patterns that are a hybrid of overlapping group sparse and sparse patterns when used as a regularizer in convex programs, and proved its theoretical convergence rates when minimizing least squares. The SOSlasso succeeds in a multi-task fMRI analysis, where it both makes better inferences and discovers more theoretically plausible brain regions that lasso and Glasso. Future work involves experimenting with different parameters for the group and l1 penalties, and using other similarity groupings, such as functional connectivity in fMRI.

## Footnotes

[1]Data and documentation available at http://www.cs.cmu.edu/afs/cs.cmu.edu/project/theo-81/www/

[2]The irregular group size compensates for voxels being larger and scanner coverage being smaller in the z-dimension (only 8 slices relative to 64 in the x- and y-dimensions).

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
