[Reviews · NeurIPS 2013]

Submitted by Assigned_Reviewer_1

This paper proposes a multi-task learning formulation using Sparse Overlapping Sets (SOS) lasso. The proposed formulation is convex and selects similar features for related learning tasks. The paper provides detailed theoretical analysis for the parameter estimation error bound. Experimental results demonstrate the effectiveness of the proposed formulation.

The SOSlasso is well motivated by multisubject fMRI studies in which functional activity is classified using brain voxels as features. The proposed formulation is inspired by the overlapping group lasso in [5]. Similar to [5], the regularizer in SOSlasso is a norm and is decomposable in a specific subspace pair. Therefore, the theoretical analysis in [9] can be extended to SOSlasso. The paper is well organized and written.

The most relevant algorithms the authors should compare are the overlapping group lasso in [5] and the dirty multi-task feature learning algorithm in [6]. It seems that the authors miss these two algorithms for comparison.
Summary: This paper proposes a multi-task learning formulation using Sparse Overlapping Sets (SOS) lasso. The paper provides detailed theoretical analysis for the parameter estimation error bound.

Submitted by Assigned_Reviewer_5

This paper presents “Sparse Overlapping Sets” Lasso (SOSlasso), a new penalty function for sparse multi-task learning (MTL), whose expression contains both the $\ell_1$ norm and the group norm penalty that allows groups to overlap. The SOSlasso is motivated by the fact that the commonly used group lasso may not be appropriate when there is uncertainty in the feature correspondence across tasks, and the proposed penalty allows one to strike a balance between two extremes: (1) lasso, where task similarity is not exploited in any way, and (2) the group lasso, which rigidly groups potentially misaligned features across tasks. By demonstrating that the proposed penalty function is a valid norm which is decomposable with respect to the model subspace of interest, the authors are able to apply the results from the work of Negahban et al. in [9] to prove error bounds for convex and differentiable loss satisfying the restricted strong convexity conditions, as well as consistency for the squared error loss. To apply the results from [9], suitable upperbounds are introduced for the SOSlasso penalty and its dual, which are used to bound the subspace compatibility constant and the dual norm of the gradient of the loss function respectively.

Overall, although the paper does not stand out to be revolutionary, it is very well written and offers a natural and valuable extension to the group lasso penalty for sparse MTL. The multi-subject fMRI experiment is an interesting application, where the imperfection in the voxel alignment during data preprocessing does indeed seem to be a good fit for the SOSlasso, as opposed to the group lasso and other existing extensions that tend to jointly select potentially misaligned voxels (features) across subjects (tasks).

The authors may want to include a short discussion on the possible extensions and limitations of the method at the end (this is easier said than done due to the page limitation, but the unnatural and abrupt ending of the paper warrants a quick fix). For example, I am curious to know how the group sizes and the degree of the overlaps were decided for the experiments. Was this part of the tuning during the cross-validation step, or are there heuristics or data-driven approach for this? How can the groups and the overlaps be set if the features are not defined over a regularly sampled (spatial) coordinate system?
Summary: The paper makes a natural and valuable extension to the group lasso penalty for sparse multitask learning, a contribution worthy for acceptance. The authors may want to include a short discussion on the limitations and possible extensions of the proposed method.

Submitted by Assigned_Reviewer_6

Comments:
This paper proposes a new method for multi-task learning where the features in different tasks are similar but not identical. The main contribution is the proposed regularizer for Sparse Overlapping Sets lasso, where the group configuration can be automatically learned. Experiments with real and synthetic data demonstrate the advantages of SOSlasso compared to the lasso and group lasso.

Overall, this paper is interesting. However, compared to earlier works, the novelty and improvements are not very clear enough. Several key problems need further clarified in this submission. In addition, connection between proposed framework and more practical applications is unclear. The paper presentation should be further improved.

Some additional comments:

1. The equation (3) is interesting point in this paper. However, it is not an easy optimization problem. It seems that w_G is already non-overlapping. Why do the variable need to be duplicated as in [5]? Another issue is how to fix the group size of each group G, for the cases with overlapping and non-overlapping respectively? The readers need know more details on how to exactly solve this problem efficiently. Without it, it is hard to evaluate its advantage.

2. The reviewer has the feeling that Section 3 contains a significant amount of existing results, while the new things offered in this section seem to be simply extended from those existing works. It is suggested to present this section in a more efficient way to be differentiated clearly from prior works.

3. The experiments are not convincing.
a) The synthetic data in Section 5.1 seems to be typical overlapping group sparse data, why does the Glasso method perform worse?
b) In Fig.2(b), when alpha = 0, all coefficients are zeros. Why the MSE of Glasso is not zero as the other two algorithms?
c) In Section 5.2, the proposed SOSlasso is compared with standard solvers lasso and Glasso. However, there is no comparison against the state-of-the-art methods of fMRI analysis and overlapped group lasso methods. Without such experiments, it is hard to validate the effectiveness of the proposed method on this application.

4. The connection between the method and fMRI application is missing. It is very difficult for readers know how to apply the proposed method on fMRI analysis, especially for those without any fMRI background. It is suggested to describe the problem in fMRI analysis briefly. The author should clarify which data is sparse or group sparse, and what is the data matrix $Phi$.

5. In table 1, the norm values of some examples are provided. Why not present the mechanism of the proposed regularizer, which can automatically select the best groups?

6. In p2 line 84, there should be a period at the end.

7. In p3 line 110-111, “Uppercase and lowercase” changes to “Lowercase and uppercase”.

8. In p3 line 119, “non zero” changes to “non-zeros”.

9. In p6 line 317, “$G \in G$” is a typo.

10. Please define MSE, AWGN etc.
Summary: This paper proposes a new method for multi-task learning where the features in different tasks are similar but not identical. The main contribution is the proposed regularizer for Sparse Overlapping Sets lasso, where the group configuration can be automatically learned. However, compared to earlier works, the novelty and improvements are not well presented and the paper should be further improved.
Author Feedback

Author rebuttal: We thank the reviewers for their detailed and insightful comments. Please find below our responses---we hope that all concerns have been addressed.

First, a general comment on how groups are defined. In the fMRI case, we used a well-known publicly released dataset in which the native resolution is lower in Z dimension compared to X and Y dimensions. Our group size of 5x5x1 sets of neighboring voxels accounts for this difference in spatial resolutions.

The group size was chosen to roughly reflect the intrinsic uncertainty in the coordinates (arising from the coordinate assignment process of [17]) and in how function is mapped anatomically from subject to subject. We did not optimize or tune the group sizes, nor did we ``cherry pick'' one particular group size that worked while others failed. The problem of optimizing the group selection process in a statistically sound manner (i.e., via CV or by incorporating a hierarchy of groups of varying sizes and shapes into our procedure) may further improve results, but this is left for future work.

For other applications, we note that our method will can be applied to any dataset that incorporates some concept of similarity among features—for instance, in a graph where edges connect similar features, cliques or well-connected subgraphs would be natural candidates for groups.

Response to Assigned_Reviewer_1

In Section 5.1, we compare our method to [5] on toy data. Note that when the groups are not sparse, [5] outperforms our method as expected. We also ran the algorithm in [5] on the fMRI data, but its results were close to chance, presumably because the neural signal actually is sparse within groups. This is discussed in more detail in the supplementary materials on fMRI. A comparison with [6] is warranted, and we will do so in future work. Since the method of [6] must, however, choose either dense groups (like Glasso) and/or unstructured sparsity (like lasso), intuitively its performance shouldn't be much better than lasso or Glasso.

Response to Assigned_Reviewer_5

We agree that the paper ends abruptly, and is notably lacking a discussion of limitations/extensions. We have reclaimed some page-space to include this in the final paper.

Please see discussion above regarding the how groups are defined.

Response to Assigned_Reviewer_6

The reviewer appears to misunderstand certain key ideas in the paper and underestimates the novelty of the theoretical work in Section 3. We hope to clarify these matters.

Non-active groups contain only zeros. Active groups contain one or more non-zero. Standard group lasso selects a subset of groups and *all* elements in each selected group are non-zero in its solution (due to noise). SOS lasso selects a subset of groups and typically sets many elements in each selected group to zero (owing to the additional L1 norm). This is precisely the sort of sparsity we are interested in; a few active groups with a few non-zeros in each active group. Standard group lasso doesn't capture this sort of sparsity well, but SOS lasso does, as can be seen from our experiments.

We respond to the points raised in your review below

1. For each group, there is a corresponding vector w_G, and the supports of these vectors overlap. The vector x is the superposition of the w_G vectors. The w_G vectors are treated as independent variables in the optimization (i.e., using the "replication" method of [5]). The optimization is straightforward and now standard [5]. See discussion above regarding the how groups are defined.

2. Section 3 shows how the general machinery in [9] can be used to analyze SOS lasso. The terminology and definitions follow [9], but all the results of Section 3 are novel and mostly non-trivial. They are not simple extensions of existing results.

3 (a) Glasso performs worse if the groups are sparsely populated because, as pointed out above, all elements in each active group are non-zero in its solution (due to noise). The extraneous non-zeros are errors, worsening its performance.

(b) The sparsest cases considered in the Figure 2(b) include at least one non-zero in each active group (i.e., alpha is slightly greater than 0), so the errors in all cases are strictly larger than zero. We will clarify this in the paper. The Glasso error is larger than lasso or SOS lasso when the groups are extremely sparse due to the issues explained in (a). If a group containing true non-zeros is not selected, then there is a bias error, otherwise the extraneous non-zeros in the Glasso solution due to noise lead to additional variance.

(c) The lasso and variants like Glasso are state-of-the-art in fMRI analysis (cf. [12], [13], [16]).

4. We have provided additional details of the fMRI experiment in the supplementary materials section. We could not include all the details in the paper due to lack of space, but we intend to do so in a longer, more detailed version of the paper. The columns of the Phi matrix are the BOLD responses at each voxel. We will clarify this in the paper.

5. Table 1 compares the lasso, Glasso, and SOS lasso penalty values for different sparsity patterns. It shows that lasso is not sensitive to the group sparsity, Glasso is not sensitive to the sparsity within groups, but SOS lasso is sensitive to both (thereby favoring the selection of a few sparsely populated groups).

We have made note of the minor corrections suggested in points 6-10, and incorporated them in the paper.